# Improved estimates on global carbon stock and carbon pools in tidal wetlands

Xiaoguang Ouyang [1]* & Shing Yip Lee [1,2]*

Tidal wetlands are global hotspots of carbon storage but errors exist with current estimates on their carbon density due to the use of factors estimated from other habitats for converting loss-on-ignition (LOI) to organic carbon (OC); and the omission of certain significant carbon pools. Here we show that the widely used conversion factor (LOI/OC = 1.724) is significantly lower than our measurements for saltmarsh sediments (1.92 ± 0.01) and oversimplifies the polynomial relationship between sediment OC and LOI for mangrove forests. Global mangrove OC stock in the top-meter sediment reaches 1.93 Pg when corrected for this bias, and is 20% lower than the previous estimates. Ecosystem carbon stock (living and dead biomass, sediment OC and inorganic carbon) is estimated at 3.7–6.2 Pg. Mangrove deforestation leads to carbon emission rates at 23.5–38.7 Tg yr$^{-1}$ after 2000. Mangrove sediment OC stock has previously been over-estimated while ecosystem carbon stock underestimated.

[1] Simon F.S. Li Marine Science Laboratory, School of Life Sciences, The Chinese University of Hong Kong, Shatin, Hong Kong SAR, China. [2] Earth System Science Programme, The Chinese University of Hong Kong, Shatin, Hong Kong SAR, China. *email: x.ouyang@cuhk.edu.hk; joesylee@cuhk.edu.hk

Tidal vegetated wetlands (mangroves and saltmarshes) are blue carbon ecosystems[1] that are highly efficient in sequestering and storing carbon for mitigating climate change[2,3]. The anaerobic sedimentary environment, high autotrophic production and ability to trap allochthonous (marine and/or riverine) sediment input[4,5] promote long-term carbon storage in these coastal ecosystems. Owing to the high carbon accumulation capacity, mangrove conservation and reforestation have been promoted in international initiatives for mitigating the risk of climate change[6]. A large number of studies have estimated sediment/ecosystem carbon stock and density in blue carbon ecosystems[7–9]. However, the current estimates on sediment carbon stock in mangrove forests are not satisfactorily constrained due to the large variation (72–936 Mg ha$^{-1}$) in individual observations and the use of conversion factors inferred from other habitats for estimating organic carbon (OC) from organic matter, without reference to potential variability associated with specific sediment types.

Loss-on-ignition (LOI) has been widely used for estimating sediment organic matter and thus a proxy for OC[10,11]. While the widely used conversion factors from organic matter to OC (e.g., 0.58 or 1.724 for OC to LOI) is commonplace in C stock estimates, these conversion factors only apply to some soils or specific components of soils, and has been found to be too low for most soils[12]. In cases where conversion factors for mangroves were estimated, the samples were collected from only one or a few sites (Supplementary Table 1). This partly, if not entirely, biases the global sediment OC stocks in mangroves when using conversion factors estimated from specific mangrove sites or other habitats to convert OC from LOI[9,13]. For example, young marshes are characterised by sediments with low OC content have an OC:LOI ratio of around 40%[14], corresponding to a LOI:OC ratio of 2.5. Low OC sediments are common in coastal ecosystems, such as calcareous deposits produced by calcifying organisms, or those with strong contributions from terrestrial and estuarine sources. Young vegetated and other coastal habitats with low sedimentary OC are usually smaller in spatial extent but are more efficient in carbon accumulation than adjacent tidal flats[15,16]. Nevertheless, their OC:LOI ratios are largely unknown, and even if measured, are not differentiated from other organic-rich sediments when estimating OC from organic matter by the LOI method[14].

Furthermore, sediment inorganic carbon (IC) is also a component of sediment carbon accumulation in blue carbon systems such as seagrasses[17]. Sediment IC deposited in coastal wetlands are associated with calcifying organisms and dissolution processes, which can have a cooling or warming effect depending on the formation process of IC (e.g., carbonate formation in seagrasses but not in chenier plains)[18,19]. Although IC is less dynamic than OC in mitigating climate change, it is an indispensable component of sediment carbon stock that has been ignored in current estimates of global sediment/ecosystem carbon stock in mangroves. Similarly, dead biomass is also a largely neglected component of the current total carbon stock in mangroves[20].

The overall sediment carbon stock is the sum of sediment OC and IC stocks, and ecosystem carbon stock consists of sediment carbon stock, living and dead biomass carbon. Thus, we also combine literature and our field measurements of IC stock to estimate sediment IC stock in mangroves, which adds to OC stock to provide a first estimate of sediment total carbon stock in global mangrove ecosystems. Ecosystem carbon stock is the sum of total sediment carbon stock, adding to literature data on living and dead biomass carbon stock.

Here, we combine data from past studies on OC and LOI and new field measurements of sediments in both mangroves and saltmarshes to better constrain the relationship between OC and LOI for different ecosystems (i.e., separately for mangroves and saltmarshes). Based on this relationship for mangroves, we then improve current estimates on global sediment OC stock in mangroves using a robust method, incorporating the median value of individual sediment OC stocks at site levels to reduce the undue influence of extreme OC stock values.

We compiled published sediment OC stock in mangroves. Our database on OC and LOI consists of 1534 observations on coupled measurements of OC and LOI (See supplementary Table 1 for the studies). The database on sediment OC stock consists of 1727 observations on sediment OC stock from 52 countries, with new data on four additional countries (Myanmar, Costa Rica, United Arab Emirates and El Salvador) compared with the previous datasets[7]. The sediment OC stock data cover around 50% of all countries that support mangroves, and these countries account for 91.9% of global mangrove area. The database on sediment IC stock consists of data on 100 sites (Supplementary Table 2). The database on dead biomass carbon consists of data on 225 sites (Supplementary Datasets), from 17% of the references on OC stock.

## Results

**Relationship between OC and LOI in mangroves and saltmarshes.** There are distinct patterns of the relationship between OC and LOI in mangrove and saltmarsh sediments. Three independent significance tests were run for the mangrove data, and the error level was set at 0.017 based on the Bonferroni correction. A strong significant polynomial relationship exists between OC and LOI for mangrove sediments ($R^2 = 0.86$, $P < 0.001$) (Fig. 1a), while a linear relationship exists between OC and LOI for saltmarshes ($R^2 = 0.99$, $P < 0.001$) (Fig. 1b). Moreover, the slope of the regression line for saltmarsh sediments (0.52 ± 0.003) is significantly lower than the constant conversion factor (OC:LOI) of 0.58 ($P < 0.001$) adopted by studies estimating OC content and then carbon stock without direct measurement of OC. The exponent (1.12) of the relationship for mangroves is significantly different from 1 ($P \ll 0.001$), i.e., a linear relationship cannot be used to approximate the polynomial relationship.

**Sediment carbon stock in global mangroves.** Our result shows that sediment OC stocks in mangroves vary with geographic, climatic and environmental factors. Sediment OC stocks in mangroves show a significant difference among different latitudinal ranges from 0 to 40° at 10° intervals (K–W $\chi^2(3) = 116.9$, $P \ll 0.001$, Fig. 2b). Further, sediment OC stock at 0–10° is significantly higher than those of all other latitudinal intervals. Similarly, sediment OC stock at 10–20° is significantly higher than those of 20–30° (M–W test, $W = 93307$, $P \ll 0.001$) and 30–40° (M–W test, $W = 46,928$, $P \ll 0.001$) but there is no significant difference between those at the latter two latitudinal ranges (M–W test, $W = 40,904$, $P = 0.15$). The higher sediment OC stock in low-latitudinal regions is consistent with their high productivity and biomass (e.g., taller tree height)[20], which provide significantly higher autochthonous carbon inputs to the sediments. In addition, sediment OC stocks in mangroves subject to high rates of relative sea level rise over the late Holocene (i.e., I–II) are significantly higher than those of other zones (M–W test, $W = 80,070$, $P \ll 0.001$, Supplementary Fig. 1a). This result is consistent with Rogers et al.[21], who found sediment carbon stocks in saltmarshes are significantly higher in zones experiencing historical relative sea level rise compare with those subject to relative sea-level stability, owing to more accommodating space in the former condition. There are also significant differences in sediment OC stocks among mangroves under different salinity

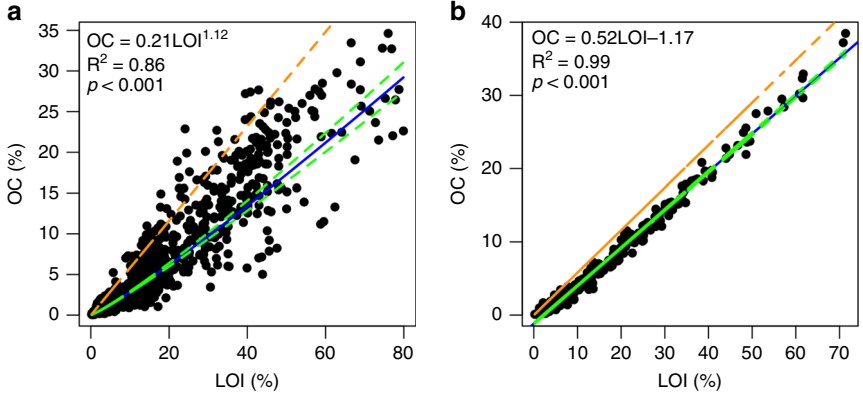

**Fig. 1 Sediment organic carbon and loss on ignition relationship. a** The relationship for mangroves. **b** The relationship for saltmarshes. The green dotted lines are the 95% confidence intervals. The brown dotted line represents the relationship of the van Bemmelen factor (OC/LOI = 0.58).

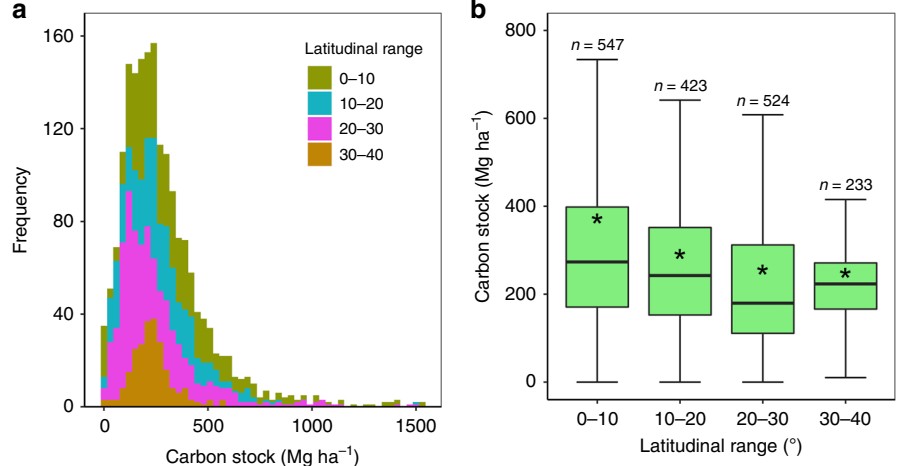

**Fig. 2 Latitudinal comparison of sediment organic carbon stocks in mangroves. a** Latitudinal frequency distribution of sediment OC stocks. **b** A whisker plot of sediment OC stocks. The stars in the whiskers show the mean sediment OC stock. The lower and upper hinges correspond to the first and third quartiles (i.e., the 25th and 75th percentiles), respectively. The lower and upper whiskers denote the lowest and highest values within 1.5 times inter-quartile ranges from the first and third quartiles, respectively.

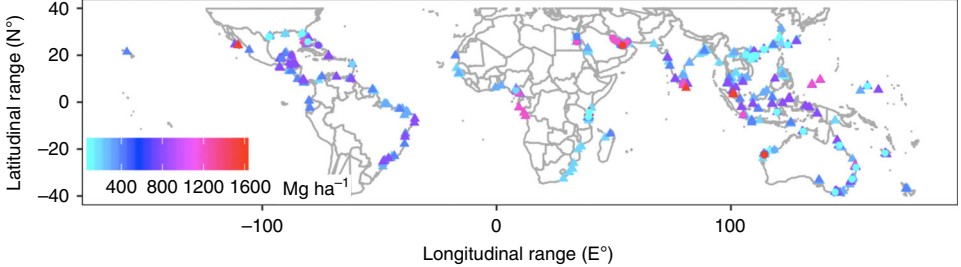

**Fig. 3 Global distribution of sediment carbon stocks in mangroves.** The triangular and circular labels denote sediment OC and IC stocks, respectively.

categories (K–W $\chi^2(2) = 23.5$, $P \ll 0.001$), forest conditions (K–W $\chi^2(3) = 6.6$, $P < 0.05$) and mangrove types (K–W $\chi^2(4) = 70.4$, $P \ll 0.001$, Supplementary Fig. 1b–d). The higher sediment OC stocks under higher salinity categories are attributed to more anaerobic conditions, which favour OC accumulation[22]. The increase in sediment OC stocks along the gradient from scrub shrub to forested conditions is also underpinned by the variation in forest productivity and biomass. The differences in sediment OC stocks among mangrove types can be explained by combined factors, including forest conditions (dwarf mangroves vs. others), salinities (lower salinities in riverine mangroves vs. high salinities

in overwash mangroves), litter and root decomposition, and the availability of organic and mineral sediments[23,24].

Sediment OC stocks in global mangroves show large variations and we used a robust method to estimate the representative OC stock. Three independent significance tests were run for all components of carbon stocks, and the error level was set at 0.017 based on the Bonferroni correction. The individual sediment OC stocks in mangroves ranged from 6.1 Mg ha$^{-1}$ (Natal coast, South Africa) to 1526 Mg ha$^{-1}$ (Gress-Rhizo-Gazi, Kenya, Fig. 3) and show large variations (mean ± standard deviation = 293.9 ± 208.25 Mg ha$^{-1}$, coefficient of variation = 70.9%). These values

**Table 1 Comparison of global mangrove sediment and ecosystem carbon stocks estimated in this study with published values.** *NA* not available; *including data from a few unpublished studies.

| Components | Number of studies | Carbon stock (Mg ha⁻¹) | Remarks | References |
|---|---|---|---|---|
| Sediment OC stock | NA | 321 (range: 272–703) | Use of mean value as the representative value of carbon stock. Synthesis of data from Kristensen et al.[5], Donato et al.[42], and Chmura et al. (2003). | Jardine and Siikamäki[25] |
| Sediment OC stock | 147* | 283 ± 193 (range: 14.9–1526.9) | Use of mean value as the representative value of carbon stock. Conversion factor for sediment OC from LOI is based on data from Palau, Indonesia. | Atwood et al.[9] |
| Sediment OC stock | 149* | 361 ± 136 (range: 86–729) | Use of mean value as the representative value of carbon stock. Conversion factor for sediment OC from LOI is based on 146 samples. | Sanderman et al.[13] |
| Total carbon stock (live biomass carbon, sediment OC) | NA | 82.8 (live biomass), 365.6 (total carbon stock) | Biomass is estimated by remote sensing. Use of sediment organic carbon estimated by Atwood et al.[9]. Total carbon stock is the sum of live biomass carbon and sediment OC stock. | Simard et al.[20] |
| Ecosystem carbon stock (live and dead biomass carbon, sediment OC and IC) | 235* (sediment OC), 25 (sediment IC), 40 (dead biomass carbon) | 237.4 (sediment OC, range: 6.1–1526.9), 34.7 (sediment IC, range: 0–1506.9), 102.5 (live biomass carbon, range: 75.8–150.3), 76 (dead biomass carbon), 450.6 (ecosystem carbon stock) | Live biomass carbon is estimated by Simard et al.[20], Tang et al.[54], Hamilton and Friess[27], Twilley, et al.[55], and Hutchison, et al.[56]. Use of median value as the representative value of carbon stock. Conversion factor for sediment OC from LOI is based on global data from 1189 samples. | This study |

are similar to those in previous studies, which also reported higher ranges of sediment OC stocks (14.9–1526.9 Mg ha⁻¹)[9] and densities (3.38–46.41 mg cm⁻³)[7] in mangroves. Both sediment OC stock and log-transformed data violate the normality assumption (Shapiro-Wilk normality test, $P \ll 0.01$). Therefore, the mean value (293.9 Mg ha⁻¹) of individual sediment OC stocks does not provide a representative value of the global sediment OC stock. Instead, the median value (237.4 Mg ha⁻¹) better represents the central value of global sediment OC stock in mangroves, and avoids the undue influence of extreme values, e.g., the very high OC stocks reported from African systems. Our estimated central value of sediment OC stock in mangroves is lower than the previous estimates of 321[25], 283[9], and 361 Mg ha⁻¹[13] (Table 1). The current estimate of global mangrove area is around 81,485 km²[26]. Combining the median value of sediment OC stock with global mangrove area, we estimated that the total OC stock in the top metre of global mangrove sediments reaches 1.93 Pg, which is 26% lower than the previous estimate of 2.6 Pg[9], and 35% lower than 2.96 Pg suggested by another study[27].

Similarly, sediment IC stock shows large variations. The individual sediment IC stocks in mangroves ranged from 0 Mg ha⁻¹ (e.g., Queensland, Australia) to 1506.9 Mg ha⁻¹ (Abu al Abyad, United Arab Emirates, Fig. 3). Again, both sediment IC stocks and log-transformed data significantly violated the normality assumption (Shapiro-Wilk normality test, $P \ll 0.017$). Therefore, the mean value (263.7 Mg ha⁻¹) does not, while the median (34.7 Mg ha⁻¹) represents the central value of global sediment IC stock in mangroves, which is 14.6% of the sediment OC stock. High sediment IC contents are confined to carbonate settings, which comprise karstic environments and Holocene reef tops[7,28]. In the carbonate settings, sediment IC stock can be as high as 549.4 Mg ha⁻¹ in karstic environments[29] and 93.4 Mg ha⁻¹ in mangroves adjacent to coral reefs[30], which triples or is one order of magnitude higher than the median value of global

sediment IC stocks. For the latter setting, sediment IC may be mostly allochthonous with coral reefs as the major source[30]. Accordingly, the mean value of global sediment IC stocks biases the representative sediment IC, since it consists of allochthonous IC, which was excluded in our estimate. Sediment carbon stock, i.e., the sum of IC and OC stock, is estimated to be 272.1 Mg ha⁻¹. The total sediment IC stock in the tope metre of mangrove sediments was estimated to reach 0.28 Pg, by combining global mangrove area and the median. By summing the total sediment OC and IC stock, the total sediment carbon stock in mangroves is estimated at 2.21 Pg.

We further estimated ecosystem carbon stock in mangroves by combining the improved total sediment carbon stock with reported living and dead biomass carbon stocks (Fig. 4). Dead aboveground biomass (litter and dead wood) and belowground biomass (dead root) carbon stock were estimated to be 16.7 and 59.3 Mg ha⁻¹ (data in Hiraishi et al.[31], Alongi[32], and others summarised in Supplementary Datasets), while living biomass carbon stock was estimated at 102.5 ± 12.3 Mg ha⁻¹ (range: 75.8–150.3 Mg ha⁻¹) from five global estimates. Adding the biomass carbon stock to our estimated sediment carbon stock, ecosystem carbon stock was estimated to be 454.5 Mg ha⁻¹ in mangroves, corresponding to the total ecosystem carbon stock of 3.7 Pg (mangrove area at 81,485 km²[26]) or 6.2 Pg (mangrove area at 137,760 km[[2]33]).

## Discussion

Our improved conversion of sediment OC stock from LOI is corroborated by studies on the relationship between OC and LOI in other biomes. The conversion factor based on our estimated slopes for saltmarsh sediments (1.92 ± 0.01) approximates the median value (1.9) for the conversion factors from compiled studies across different biomes[12]. Other studies also established

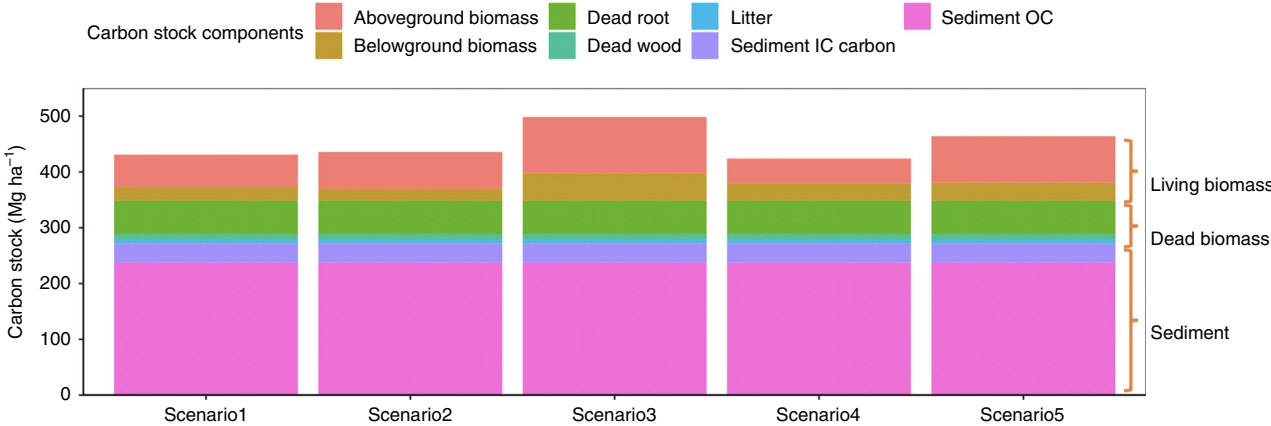

**Fig. 4** Global ecosystem carbon stocks for mangroves. Living biomass (aboveground and belowground biomass) carbon stocks in the five scenarios are cited from Simard et al.[20], Tang et al.[54], Hamilton and Friess[27], Twilley et al.[55], and Hutchison et al.[56], while other components of carbon stock come from the same source. The unit of carbon stock is Mg ha$^{-1}$.

relationships between OC and LOI for regional coastal wetlands (Supplementary Table 3). Generally, the equations based on their relationships also suggest that the conversion factor of 0.58 is too high for coastal wetlands. The conventional conversion factor of 0.58 therefore grossly overestimated saltmarsh OC content and biased mangrove OC content, which has a polynomial relationship with LOI, and thus OC stock. Furthermore, the relationships reported in this study are based on data from both the Indo-west-Pacific and Atlantic-east-Pacific mangrove biogeographic regions, while the data of past studies focused on specific sites in either of the biogeographic regions. In coastal wetlands, we also found different relationships for mangroves and saltmarshes. This difference begs questions on the underlying biological mechanisms. Mangroves consist of both trees and shrubs, while saltmarshes consist of grasses and shrubs. Woody plants have higher C content in all tissues compared with herbs and shrubs[34]. Mangrove trees have woody tissues with more structural carbon while grasses and shrubs have soft tissues with less structural carbon. From our field measurements, we found sediments of forest (tall, 7 m) mangroves (i.e., Mai Po) are characterised with high LOI while the scrub (2 m) mangroves (i.e., Wu Shek Kok and others) are characterised with low LOI (Supplement Dataset on LOI). The mangrove sediments with high and low LOI have different conversion factors (Supplementary Notes). This could explain the polynomial relationship for mangroves and a linear relationship for saltmarshes. Accordingly, our findings improve past relationships developed for regional coastal wetlands and highlight the difference in the OC:LOI relationships between mangrove and saltmarsh ecosystems.

Our results highlight that mangrove sediment OC stock has previously been over-estimated while ecosystem carbon stock underestimated due to the use of inappropriate conversion factors and ignoring dead biomass and sediment inorganic carbon. The previous studies used the mean value of individual sediment OC stock as the representative carbon stock and/or a smaller database[25] with only up to 54% of our collected observations. The high individual carbon stocks (1000–2000 Mg ha$^{-1}$) are an order of magnitude higher than the mean value (200–300 Mg ha$^{-1}$) of the carbon stocks, which led to overestimates of the global mangrove carbon stock. The smaller database missed a large number of carbon stock data hotspots for mangroves, e.g., Indonesia, thus also reducing the validity of the projected estimate. Further, past global syntheses improve the estimate of sediment OC from OM using conversion factors developed from samples at specific mangrove sites[9] or other habitats other than

mangroves[13] (Supplementary Table 3). For the latter group of studies, a conversion factor was estimated based on a small database (146 samples) but not used in their further analysis. Our estimate on ecosystem carbon stock is higher than the previous estimate of 365.6 Mg ha$^{-1}$[20] (Table 1), which neglects the dead biomass component. There are other estimates of global mangrove area (e.g., 137,600 km$^2$ in 2010[35]), which may result in the variation of total mangrove ecosystem carbon stock. Nonetheless, mangrove area is a poor constraint on carbon densities or other mangrove functions[36], as canopy height[20] and species identity[9,37] could also result in large variation in carbon densities, as do salinity categories, forest conditions and mangrove types, as found in this study. Sediments account for the largest proportion (60.4%) of ecosystem carbon stock. Mangrove sediments are not carbon saturated and can accrete considerable mineral and organic carbon input from marine and/or riverine sources over millennia[38,39], and the anoxic saline conditions reduces organic matter and root decomposition, which also contribute to sediment carbon storage and surface elevation increase[24,40] through biogenic accretion[41]. We only extrapolate data on sediment carbon stock in the top metre of sediment but mangrove soils can be much deeper (>3 m in many estuarine settings)[42]. Therefore, our estimate on sediment carbon stock based on the top metre is an underestimate, as the deposits can be several-fold deeper and we only consider the layer that is most prone to remineralisation.

Climate and anthropogenic activities are two important drivers of changes in mangrove ecosystem carbon stocks but the mechanisms of their influence are fundamentally different. The variation of sediment OC and belowground but not aboveground biomass carbon in mangroves is associated with annual rainfall which may underpin longer sediment inundation period and hinder organic matter decomposition[43]. Therefore, it is necessary to consider all components of ecosystem carbon stocks when estimating potential changes in mangrove ecosystem carbon derived from changing rainfall patterns. Mangroves have been lost at an alarming rate due to deforestation activities, e.g., transformation to aquaculture ponds, paddy and oil palm fields[44,45]. Previous studies have evaluated $CO_2$ emissions resulting from anthropogenic disturbances (e.g., mangrove deforestation) but only living biomass and/or sediment OC were included in their assessment. We further estimated $CO_2$ emissions from mangrove clearing after 2000 by including our estimated ecosystem carbon stock. Assuming the total carbon loss across all carbon pools being similar to previous estimates[27,46,47] and an annual loss rate of 0.17%[27], this results in the global

carbon loss rate of 6.4 Tg year$^{-1}$ (mangrove area = 83,495 km$^2$) to 10.6 Tg year$^{-1}$ (mangrove area = 137,760 km$^2$). This corresponds to a $CO_2$ emission rate of 23.5–38.7 Tg year$^{-1}$. Our lower estimate of $CO_2$ emission rate was 3.8% lower than the estimate of previous study (24.38 Tg year$^{-1}$) based on the same mangrove area in 2000[27], owing to the different baseline dataset being used, e.g., different sediment OC stocks.

## Materials and methods

**Literature data**. Sediment loss on ignition (LOI) in mangroves and saltmarshes and sediment carbon stock in mangroves were collected from literature search in http://www.sciencedirect.com/ and http://pcs.webofknowledge.com/. Sediment LOI data search was based performed using loss on ignition combined with either mangrove or saltmarsh OR salt marsh in Abstract, title and Keywords or Topic, title. Sediment carbon stock was searched using carbon combined with mangrove in the same way. We sifted through the results for studies reporting both sediment organic carbon (OC) and organic matter estimated by LOI, as well as sediment carbon stock in mangroves. We found 30 studies reporting both sediment OC and LOI in mangroves and saltmarshes (Supplementary Table 1), and 235 studies reporting sediment carbon storage in mangroves (Supplementary Dataset on sediment OC). Among the 235 studies, the data in 22 studies were used to estimate carbon content from LOI using the van Bemmelen factor (OC/LOI = 0.58) or conversion factors developed from limited mangrove sediment samples[9] or other habitats rather than mangroves[13]. Sediment IC density in mangroves was synthesised in a recent review[17]. Based on this database and other studies (Supplementary Table 2), we estimated the IC stock in the top one metre of mangrove sediments.

We also collect the hierarchical set of site, core and depth information of the sediment OC stock dataset, along with salinity, forest conditions, Holocene relative sea level rise zones and mangrove types (Supplementary Dataset on site, core and depth). Based on the Thalassic series, salinity codes of the samples were categorised into oligohaline (0.5–5 ppt), mesohaline (5–18 ppt), polyhaline (18–30 ppt) and mixoeuhaline or more saline groups (>30 ppt). Based on the definition of mangrove types[48], Mangrove types were sorted into fringe, riverine, interior, dwarf and overwash mangroves. We use the interior type to represent the basin and hammock types in the original definition due to limited information allowing us to differentiate them. The sampling locations were also divided into high rates of relative sea level rise over the late Holocene zones (i.e., I, II) and others (III, IV, and V) based on the description of Rogers et al.[21]. Forest conditions were divided into shrub (dominated by shrubs <5 m height), forested (dominated by mature trees >5 m) and forested to shrubs (dominated by both shrubs and mature trees) based on the description of Smithsonian Environmental Research Centre (https://serc.si.edu/coastalcarbon/database-structure).

**Field data**. Furthermore, we supplemented data on LOI and IC stock from our field sampling campaign which add to sediment IC dataset and the dataset examining the relationship between sediment LOI and OC. Sediment samples were collected from mangroves and saltmarshes at Ting Kok (22°28′N, 114°13′E), Yung Shue O (22°25′N, 114°16′E), Wu Shek Kok (22°32′N, 114°13′E) and Mai Po Nature Reserve (22°30′N, 114°02′E), Hong Kong (Fig. 1) during April–October 2018. These locations represent a continuum along the estuarine–oceanic transition. Sediment samples were collected by a corer to the depth of 1 m at Mai Po Nature Reserve and 20–40 cm at other sites, depending on the ease of penetration of the sediments. The samples were sliced at 5-cm intervals for the depth 0–30 cm, 10-cm depth interval for the depth 30–60 cm and 20 cm for the rest. All the samples were kept under 4 °C in a cooler box in the field, and transferred to a freezer (−20 °C) in the laboratory. Within 3 days after collection, the samples were heated at 60 °C to a constant weight in an oven. Sediment bulk density was estimated as the dry weight divided by the volume. The sediment samples were ground, and passed through a 2-mm mesh. The samples were analysed for organic matter content by the loss-on-ignition method. Generally, sediment (<1 g) samples were heated at 550 °C in a muffle furnace (Linberg/MPH, USA) for 4 h, and the difference in weight before and after combustion is taken as an estimate for the organic matter content of the samples. Aliquots of all the samples (around 2 mg) were also collected before and after LOI analysis and stored in tin capsules, and analysed for carbon and nitrogen content using an Elemental Analyser (Perkin-Elmer II CHNS/O, UK). The carbon contents of samples collected before and after LOI analysis are the estimates of total carbon and IC, and OC is the difference between total carbon and IC. The sediment samples after LOI analysis at 550 °C were further heated at 950 °C in the muffle furnace for 2 h. Assuming a molecular weight of 60 g carbonate ($CO_3^{2-}$) and 44 g for $CO_2$, the weight loss of LOI at 950 °C multiplied by 1.36 is the theoretical weight of the carbonate in the sediment sample. Sediment IC stock was sediment IC content multiplied by the bulk density. Further explanations on the method can be found in Supplementary Notes.

**Data analysis**. Linear regression was used to examine the relationship between sediment OC content and LOI. Before regression analyses, the assumption of normality was checked by the Shapiro-Wilk normality test ($\alpha = 0.05$). When the hypothesis of normality cannot be met, data were log-transformed. The individual slope was also compared with the commonly adopted conversion factor (i.e., LOI/OC = 1.724 or OC/LOI = 0.58) used in literature by the Wald test as described by Zar[49]. The exponent of the relationship between OC and LOI in mangroves was compared with 1 in the same way. The differences in sediment OC stock at latitudinal intervals were assessed using the Kruskal Wallis rank sum test followed by Mann–Whitney $U$-tests.

The estimate of global mangrove OC stock was improved by a bottom-up method. First, the estimate of individual OC stock was improved by the above relationships between LOI and OC when individual studies estimated sediment OC from LOI with constant conversion factors. Global mangrove OC stock was estimated as the representative central values of the individual improved estimates on mangrove OC stocks. The individual values of mangrove OC stock were checked for normality using the Shapiro-Wilk normality test. When raw or transformed data (e.g., log-transformed) did not meet the normality assumption, the median of global mangrove OC stock was reported instead. When data met the normality assumption, the mean of individual mangrove OC stock was used. Otherwise, the geometric mean was used when the transformed data showed a normal distribution. Global mangrove IC stock was estimated using the same method. For all the statistical tests, the Bonferroni correction was applied to adjust the type I error to $\alpha' = \alpha/n$, where $\alpha = 0.05$ and $n$ is the number of tests)[50] to account for the effect of multiple tests.

Global ecosystem carbon stock in mangroves is estimated by summing up sediment, living and dead biomass carbon stock. There are a series of studies estimating mangrove living biomass and/or carbon stock using either remote sensing or synthesised data on field surveys but based on different mangrove area in 2000, 2012, or earlier. Thus, we extracted the data by using the unit-areal biomass carbon (Mg ha$^{-1}$) rather than total biomass carbon to estimate the ecosystem carbon stock. When only living biomass was available, we estimated the above-ground and below-ground biomass by multiplying a conversion factor of 0.451[31] and 0.39[51], respectively. The ecosystem carbon stocks were estimated under five scenarios based on the different studies on global living biomass carbon stocks: 82.8 Mg ha$^{-1}$ (scenario 1), 87.7 Mg ha$^{-1}$ (scenario 2), 150.3 Mg ha$^{-1}$ (scenario 3), 75.8 Mg ha$^{-1}$ (scenario 4), and 116 Mg ha$^{-1}$ (scenario 5).

Data are presented as mean ± standard error (SE). All data analyses were conducted via R language[52]. R package car[53] was used to conduct the Wald test.

## Data availability

The authors declare that the data supporting the findings of this study are available within the paper and its supplementary information and dataset files. Data are also available upon reasonable request sent to Xiaoguang Ouyang.

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

## Acknowledgements

We thank Professor Michael I. Bird at James Cook University, Australia, for providing his data on both sediment loss-on-ignition and organic carbon. X.O. is supported by a postdoctoral fellowship from The Chinese University of Hong Kong. This study was supported by a Direct Grant BL-17922 from CUHK.

## Author contributions

X.O. and S.Y.L. designed this study. X.O. conducted sediment sample collection and analysis, as well as data analysis. X.O. and S.Y.L. contributed equally to writing the manuscript.

## Competing interests

The authors declare no competing interests.

## Additional information

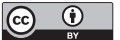

