## [Peer Review File · Nature Communications]

Reviewers' comments:

Reviewer #1 (Remarks to the Author):

This paper uses a range of published and new data to examine the relationship between organic carbon estimated using loss on ignition (LOI) methods, which are often used to estimate organic carbon because the measurements are inexpensive to perform, and those measured using combustion (e.g. using an elemental analyser). They find strong relationships (as shown before), but which have different slopes to those reported earlier in the literature, reflecting the growing number of studies from a wider range of study sites. They use these relationships (although they use two different equations for mangroves, for Low and high OC sites, which is not clear in the main text, but appears in the supplementary) to re-estimate the OC stocks in mangroves, concluding stocks have been overestimated, although emissions are similar (within 3%) to other past studies.

There are two other components of the study: estimation of inorganic carbon stocks in mangroves and carbon stocks in woody debris, which seem to add-on's to the main text.

Overall, the manuscript makes useful contributions to the field, but it seems a collection of issues rather than a coherent piece of research.

More specific points

The statement that there is a "universal conversion factor" suggests that all blue carbon stock assessments use this factor when this is not the case. Additionally, there is no reference provided for this statement.

That mangrove carbon stock assessments "ignore" inorganic carbon stocks is overstated. As the data set presented shows, in most mangrove ecosystems inorganic carbon is very low and may not be reported because preliminary measures find it is low (usually reported in methods), unless it is of particularly of interest (e.g. see Xiong et al. 2019, Biology Letters).

"Young sediments" – are these sediments recently colonized by intertidal plant communities?

This statement: "Nevertheless, their OC:LOI ratios are largely unknown, and even if measured, are not differentiated from other organic-rich sediments when estimating OC from organic matter by the LOI method." Is rather sweeping with no reference to indicate where this is the case.

The statement: "incorporating the representative values of individual sediment OC stocks." meaning was not clear.

The polynomial term for mangrove LOI-OC was not significant, so the relationship should not be reported as polynomial.

Atwood et al. 2017 also reports a higher OC stock in tropical locations, but this is not discussed.

Sanderman et al. 2018 is not cited, yet this is a global map of soil OC stocks in mangroves. Rovai et al. 2018 also provide a global analysis (and map) of OC density that could be compared.

In the data set it would be good to identify which lines are included in prior compilations and which are not (new data).

Sanders et al. (L&O Letters) suggests that OC stocks in mangroves correlates with rainfall, which may underlie both productivity and soil water content.

The authors point out that using means values to estimate global OC stocks may overestimate and use of medians would be more conservative. To make this point more clearly the authors should calculate values based on means and medians so readers can view the magnitude of this effect.

The area estimates of mangroves vary and are responsible for much of the variation in prior estimates of total mangrove OC stocks and emissions (see Atwood et al. 2017). The authors consider this point, but the discussion of this point could be developed further. Additionally, there are new estimates of global mangrove cover by Global Mangrove Watch 2015 and 2017, which should be considered.

The estimate of IC stocks for mangroves uses the geometric mean, but this likely overestimates stocks because high levels of IC are likely confined to particular geomorphic settings. Twilley et al. (Frontiers Ecol Evol) and Rovai et al. (Nat Geosci) discuss this, as do Adame et al. (2013 Plos One) and there are new data in Xiong et al. (Biol Letters) which suggest most of the IC in the site is

allochthonous (from adjacent reefal habitats).

Reviewer #2 (Remarks to the Author):

This paper presents summary statistics including corrected site-level median values of carbon stocks for global mangroves. There are three parts of the argument: 1. People are over-reporting organic carbon because they are relying on the van Bemmelen factor (0.58 gOC per gOM) as a conversion factor, 2. people are underreporting total stocks because they neglect carbonates, and 3. people are underreporting above-ground stock because they neglect dead wood.

This first aspect of the study, in my opinion, is not as novel as the authors claim it is. Authors say people have been relying on 0.58 van Bemmelen factor. I have two issues with this line of questioning. First, I don't think they ever really justified that this is a problem in the literature. Nowhere does it say how many studies in the review used this factor vs the Craft 1991 regression or another study-specific regression. Second, It's widely known that the vanBemmelen factor is too high for wetland soils. Several studies in the last year have identified and handled this in various ways providing alternative, but oftentimes very similar OM to OC equations, that do not deviate that much from the Craft 1991 regression some cited in text, others not (Holmquist 2018; Sanderman 2018). I think it's good to have this reinforced in the literature, and I agree that the van Bemmelen factor needs to go, but I don't think this is an innovative aspect of the paper.

The second argument, that underreporting carbonate stock leads to an underreporting of total stock. This may be true; I think this is missing important context. The process of forming and trapping of carbonates can have a net-warming or cooling effect depending on where carbonates were generated and what the scope of the analysis is (Howard; Lu). In some cases, long-term carbonate formation reverses cooling trend of organic carbon burial. This is an essential piece of context on the connection between formation rates, observed stocks, and net greenhouse gas balance that I did not see anywhere in the manuscript.

The third argument, that stocks are underreported because they do not include dead wood I think was also a bit weak. I never saw any citations or summary statistics indicating how many studies lacked dead wood, or how many included it. I know that there is guidance for accounting for dead wood in the IPCC wetland supplement guidance. I can also think of a few studies off the top of my head that do include it, although they tend to be tidal freshwater forested literature (Magonigal 1997; Krauss et al., 2018). Maybe there's a hole in the mangrove literature I'm not aware of that could be communicated better than answered by this point.

If the editors decide to seek a revised manuscript, I have some suggestions below.

Overall.

1. The authors claim that saltmarshes and mangroves have different OM to OC ratios. If they think this is the result of a biological mechanism, I would like this explored more in the discussion. Also, could be that this is an artifact of the data synthesis. This could be addressed by trying a mixed-effects model, the same model as is done here with a 'study level' random effect.
2. Figures – Figure 1. The log transformation makes the 95% CI's look like their exactly on the median line. There is missing symbology in the equations. The images are unnecessary. The top figure has log-transformed axes and the bottom they are linear. This is misleading and implies that the two relationships have more similar fits than they actually do. I recommend putting them on the same comparable axes and adding the 0.58 gOC per gOM line in for comparison.
3. I suggest a more thoughtful use of color. The color schemes for figures 2 and 4 are the same, but

the colors mean different things. I suggest changing the palates in one or both.

4. The histogram in fig. 2 doesn't really communicate the fact that there are significant differences in carbon stocks between latitudinal clusters to me. The variance in carbon stocks at different latitudinal gradients seems interesting and maybe deserves more attention in the main text.

5. SI table – There is a lot of unpublished data listed, and I would like to see any personal communications or submitter contact information cited in SI.

6. I suggest a column indicating the precision (number of decimal points) appropriate for the lat-long's.

7. I suggest machine-readable and analysis-ready datasets for everything they've produced in house (Wilson et al 2017; Culina et al 2018). I also suggest adopting the hierarchical set of site, core, depth series table format recommended by the Coastal Carbon Research Coordination Network (<https://serc.si.edu/coastalcarbon/database-structure#site-level>).

Line comments.

Line 62: Is the site, the core, or the depth interval the fundamental unit of the analysis? I think depth interval is best.

Line 71: I'd prefer a more precise or quantitative phrase than 'clear cut'.

Line 73: Somewhere in the results section it would be good to say how many significance tests were run, how independent they were. Consider running a Bonferroni correction to p-values to account for multiple testing.

Line 76-78: The claim is that 0.58 gOC per gOM is adopted by most studies. I haven't seen any evidence for this in the paper. In the SI it seems like a small subset of 13 studies.

Also when reporting the regression equations, the authors should add the standard error of the parameter estimates.

I think something more rigorous is needed than dropping a parameter because it is not significantly different from one. I suggest comparing a linear and polynomial versions of the model fit separately using Akaike's Information Criterion.

Were salt marsh and mangrove relationships fit separately, or were they fit using the same model and an ecosystem type fixed effect?

Line 80-89: How does this compare with Rogers et al 2019, who found that carbon stocks were higher in the regions where there was more historical isostatic sea-level rise forming accommodation space?

Line 96: Would this still be significant after adjusting for multiple testing?

Line 102: I think this sentence should be in the abstract.

Was it the correction of past studies using the 0.58 gOC per gOM or the switch from reporting means to medians which caused the 24% drop in total estimated stocks?

Materials and methods – I could use more details regarding the scenarios depicted in Fig. 4. I did not see them referenced in the main text at all and when I finished reading the paper I did not have a good idea as to where they came from and how they fit into the research questions.

Culina, A. et al., 2018. Navigating the unfolding open data landscape in ecology and evolution. *Nature Ecology & Evolution*, pp.1–7.

Holmquist, James R., Lisamarie Windham-Myers, Norman Bliss, Stephen Crooks, James T. Morris, J. Patrick Megonigal, Tiffany Troxler et al. "Accuracy and precision of tidal wetland soil carbon mapping in the conterminous United States." *Scientific reports* 8, no. 1 (2018): 9478.

Howard, Jason L., Joel C. Creed, Mariana VP Aguiar, and James W. Fourqurean. "CO₂ released by carbonate sediment production in some coastal areas may offset the benefits of seagrass "Blue Carbon" storage." *Limnology and Oceanography* 63, no. 1 (2018): 160-172.

IPCC. 2013 Supplement to the 2006 IPCC Guidelines for National Greenhouse Gas Inventories: Wetlands. (IPCC, Switzerland, 2014).

Krauss, Ken W., Gregory B. Noe, Jamie A. Duberstein, William H. Conner, Camille L. Stagg, Nicole Cormier, Miriam C. Jones et al. "The role of the upper tidal estuary in wetland blue carbon storage and flux." *Global Biogeochemical Cycles* 32, no. 5 (2018): 817-839.

Lu, Weizhi, Chang'an Liu, Yue Zhang, Caifen Yu, Pifu Cong, Junsheng Ma, and Jingfeng Xiao. "Carbon fluxes and stocks in a carbonate-rich chenier plain." *Agricultural and Forest Meteorology* 275 (2019): 159-169.

Megonigal, J. Patrick, William H. Conner, Steven Kroeger, and Rebecca R. Sharitz. "Aboveground production in southeastern floodplain forests: a test of the subsidy-stress hypothesis." *Ecology* 78, no. 2 (1997): 370-384.

Rogers, Kerrylee, Jeffrey J. Kelleway, Neil Saintilan, J. Patrick Megonigal, Janine B. Adams, James R. Holmquist, Meng Lu et al. "Wetland carbon storage controlled by millennial-scale variation in relative sea-level rise." *Nature* 567, no. 7746 (2019): 91.

Sanderman, Jonathan, Tomislav Hengl, Greg Fiske, Kylene Solvik, Maria Fernanda Adame, Lisa Benson, Jacob J. Bukoski et al. "A global map of mangrove forest soil carbon at 30 m spatial resolution." *Environmental Research Letters* 13, no. 5 (2018): 055002.

Wilson, Greg, Jennifer Bryan, Karen Cranston, Justin Kitzes, Lex Nederbragt, and Tracy K. Teal. "Good enough practices in scientific computing." *PLoS computational biology* 13, no. 6 (2017): e1005510.

Reviewer #3 (Remarks to the Author):

Overall I found the manuscript to contain material that would be a worthwhile contribution to the literature with several innovative elements. My criticisms are broad in nature, so I have not provided line-by-line comments. My primary criticism is that it was unclear whether the goal of this manuscript was to critique and suggest improvements for factors to convert loss-on-ignition (LOI) to organic carbon (OC) or to provide an updated global mangrove C stock estimate. In its current form, the manuscript attempts to do both, but the combination of these separate goals created many weaknesses and elements of confusion. As one example, it was confusing whether the updated global C stock estimates were a result of the different conversion factors, or the new data collected, or other reasons. I recommend splitting the manuscript into two. A manuscript that were to establish improved conversion factors that potentially could be used throughout the scientific community would be a highly valuable contribution to the literature and could include an implications section on how this may affect global stock estimates. A manuscript on new global-stock mangrove carbon estimates, using these new conversion factors and other new data, would also be a highly valuable contribution to the literature. Both of these contributions are sufficiently important in scope to warrant their own publication and a more comprehensive and focused treatment.

For the conversion factors paper, I suggest adding:

- A clear and comprehensive review of conversion factors that have been developed in the literature.
- A similar review of how these conversion factors have been applied. A table summarizing these uses would be very helpful. This could be restricted to global C stock estimates, but could also be highly valuable if it covers C stock estimates at other scales. The use of conversion factors is far broader in scope than just for global C stock estimates.

For global mangrove C carbon stock estimates, I suggest adding:

- A clear and comprehensive review of previous global mangrove C stock estimates and how and why they differ from the present study. A table would be useful.

Additional broad criticisms:

- The manuscript focuses on the role of coastal wetlands in mitigating climate change, but then criticizes previous studies for not including inorganic C in C stock estimates. Inorganic C is of course a component of the total C pool of a wetland, but is far less dynamic than organic C and probably plays little role in the ecosystem service of greenhouse gas mitigation. For this reason, it seems best to separate these pools when doing C stock estimates as a means to understand greenhouse gas mitigation.
- Also related to greenhouse gas mitigation is the issue of human alterations. There is a substantial difference in terms of understanding climate forcing in quantifying C stocks and quantify C stock changes resulting from human activities. This separation is generally not made clear in the manuscript.
- The carbon stock estimate was only for mangroves, but the conversion factors were for both mangroves and marshes, which was confusing and not made clear in the paper. This is another reason to split the paper.
- There are quite a few comments such as the one on line 35 "but has been ignored in current estimates" without references. See lines 39–40 and 76 as other examples, but there are others as well. This would be remedied by have a more comprehensive analysis of conversion factors in the literature.
- The manuscript is written as if the 1.724 LOI/OC ratio is essentially the only one that has been used in the literature, but this is incorrect – there are many examples of other ratios that have been used. In fact, in a recent important publication this isn't the ratio that is suggested for use in mangroves (see Table 3.7 in Fourqurean et al. - 2014 - Coastal Blue Carbon Methods for Assessing Carbon). The authors may also consider if there is pertinent information in the recently published book A Blue Carbon Primer: The State of Coastal Wetland Carbon Science, Practice and Policy.
- The authors state on line 77 that the polynomial relationship was not statistically significant, but do not explain/defend why it was then used.
- The manuscript is written as if the new data that were collected were global in nature, but only in the methods is it clarified that all of these data come from China (I believe). There is nothing wrong with adding data from China to this global database, but it should be explicit throughout the manuscript. It should also be discussed about whether these data fill a data gap—was China and/or these wetland types a data gap? Is this new database in any way weighted with the large number of samples from China?
- The use of the term "correct" should be replaced with something like "improve". As scientists, we will never have the "correct" answer for a statistical population.

I should note that although I have expertise in wetland soil carbon cycling, I don't have expertise in global-scale carbon stock estimation, so I was not able to critique this element of the manuscript.

Reviewer #1		
Overall comments		
They find strong relationships (as shown before), but which have different slopes to those reported earlier in the literature, reflecting the growing number of studies from a wider range of study sites. They use these relationships (although they use two different equations for mangroves, for Low and high OC sites, which is not clear in the main text, but appears in the supplementary) to re-estimate the OC stocks in mangroves	The previous published estimates on the relationship were either studies on specific sites or global analyses of database with limited coverage (Supplementary Information Table 3). We have developed relationships based on an expanded database from our synthesis and additional original field measurements, which cover both the Indo-west-Pacific and Atlantic-east-Pacific biogeographic regions . With growing interest in blue carbon, our evaluation of the conversion factor is timely as well as important. We presented a revised and different relationship between OC and LOI based on all collated mangrove data in the main text of the manuscript as this is critical to improving future estimates of OC from LOI. We further explain why we developed the relationships for low and high LOI sediments in the Supplementary Information.	Supplementary Information Table 3, l. 186-188, Supplementary Information
Specific comments		
1. The statement that there is a “universal conversion factor” suggests that all blue carbon stock assessments use this factor when this is not the case. Additionally, there is no reference provided for this statement.	We have revised this sentence to avoid overstating the use of a single conversion factor.	l. 10, l. 34, l. 37
2. That mangrove carbon stock assessments “ignore” inorganic carbon stocks is overstated. As the data set presented shows, in most mangrove ecosystems inorganic carbon is very low and may not be reported because preliminary measures find it is low (usually reported in methods), unless it is of particular interest (e.g. see Xiong et al. 2019, Biology Letters).	Sediment inorganic carbon is assumed to be less important in many references. However, we found globally sediment IC contributes 14.6% of sediment OC stock, and therefore include it as a component of sediment carbon stocks in our discussion, which is another contribution of our study. Actually, sediment	l. 151-157, l. 54-57

	IC can have a warming or cooling effect depending on the formation process (Refer to our later response to Reviewer #2's comment on argument 2).	
3. "Young sediments" – are these sediments recently colonized by intertidal plant communities?	The reference did not explain this point and we have rephrased the sentence to avoid confusion.	l. 44-46
4. This statement: "Nevertheless, their OC:LOI ratios are largely unknown, and even if measured, are not differentiated from other organic-rich sediments when estimating OC from organic matter by the LOI method." Is rather sweeping with no reference to indicate where this is the case.	We have added a reference related to this issue in the revised manuscript.	l. 52
5. The statement: "incorporating the representative values of individual sediment OC stocks." meaning was not clear.	We have revised this sentence to improve clarity	l. 66-67
6. The polynomial term for mangrove LOI-OC was not significant, so the relationship should not be reported as polynomial.	In the first version of the manuscript, we stated that 'the exponent (1.02) of the polynomial relationship was insignificantly different from 1' which does not mean the polynomial model for mangrove LOI-OC was not significant. It just means the polynomial relationship can be approximated by the linear relationship. In the revised ms, we have improved the relationship with more data and now the exponent is different from 1, i.e. the polynomial relationship cannot be approximated by a linear relationship.	l. 96-98
7. Atwood et al. 2017 also reports a higher OC stock in tropical locations, but this is not discussed. Sanderman et al. 2018 is not cited, yet this is a global map of soil OC stocks in mangroves. Rovai et al. 2018 also provide a global analysis (and map) of OC density that could be compared.	We have cited one more reference (as the reviewer suggested) and discussed the results with more reference to the other two papers, which are already cited in the ms.	l. 133-135, l. 140-142
8. In the data set it would be good to identify which lines are included in prior compilations and which are not (new data).	We have added a column in the data set to show whether the data are from prior compilations or not.	Supplementary Dataset on sediment OC

9. Sanders et al. (L&O Letters) suggests that OC stocks in mangroves correlates with rainfall, which may underlie both productivity and soil water content.	We have cited this reference in the discussion of the impact of climate forces on sediment OC stocks.	l. 231-235
10. The authors point out that using mean values to estimate global OC stocks may overestimate and use of medians would be more conservative. To make this point more clearly the authors should calculate values based on means and medians so readers can view the magnitude of this effect.	We have added the values of sediment OC stocks calculated based on means in addition to those based on the medians.	l. 132
11. The area estimates of mangroves vary and are responsible for much of the variation in prior estimates of total mangrove OC stocks and emissions (see Atwood et al. 2017). The authors consider this point, but the discussion of this point could be developed further. Additionally, there are new estimates of global mangrove cover by Global Mangrove Watch 2015 and 2017, which should be considered.	We have further discussed the variation of total mangrove OC stock due to the area estimates, with reference to new estimates of global mangrove cover. However, mangrove area is not a good proxy for carbon densities since tree height and species identity may also result in large variation in carbon densities. A sentence and additional references are added to highlight this point.	l. 215-221
12. The estimate of IC stocks for mangroves uses the geometric mean, but this likely overestimates stocks because high levels of IC are likely confined to particular geomorphic settings. Twilley et al. (Frontiers Ecol Evol) and Rovai et al. (Nat Geosci) discuss this, as do Adame et al. (2013 Plos One) and there are new data in Xiong et al. (Biol Letters) which suggest most of the IC in the site is allochthonous (from adjacent reefal habitats).	With more data on sediment IC, we found that the median value is representative of global mangrove sediment IC stocks, and replaced the geometric mean in the ms. We also provide the mean value of sediment IC stocks. We have discussed the likely high IC in karstic environments and Holocene reef tops, citing the relevant references.	l. 151-153
Reviewer #2		
Comments on arguments		
1. Authors say people have been relying on 0.58 van Bemmelen. factor. I have two issues with this line of questioning. First, I don't think they ever really justified that this is a problem in the literature. Nowhere does it say how many studies in the review used this factor vs the Craft 1991 regression or another study-specific regression. Second, It's	We have clarified how many studies used the conversion factor vs. the relationships developed specifically for coastal wetlands. We have listed and compared the developed relationships between OC and LOI, and stressed the necessity to develop a	l. 262, Supplementary Information Table 3, l. 180-201

widely known that the vanBemmelen factor is too high for wetland soils. Several studies in the last year have identified and handled this in various ways providing alternative, but oftentimes very similar OM to OC equations, that do not deviate that much from the Craft 1991 regression some cited in text, others not (Holmquist 2018; Sanderman 2018). I think it's good to have this reinforced in the literature.	global relationship using available data from past studies and our new field data. We agree that the van Baemmelen factor (OC:LOI=0.58) is too high for wetland sediments while Craft (1991) was developed from marsh sediments from a few sites in USA. The new relationship we developed is an advancement over the past relationships because we include (1) separate relationships for mangroves and saltmarshes; (2) data from coastal wetlands covering both the Indo-west-Pacific and Atlantic-east-Pacific biogeographic regions.	
2. The second argument, that underreporting carbonate stock leads to an underreporting of total stock. This may be true; I think this is missing important context. The process of forming and trapping of carbonates can have a net-warming or cooling effect depending on where carbonates were generated and what the scope of the analysis is (Howard; Lu). In some cases, long-term carbonate formation reverses cooling trend of organic carbon burial. This is an essential piece of context on the connection between formation rates, observed stocks, and net greenhouse gas balance that I did not see anywhere in the manuscript.	We have briefly clarified the biogeochemical process of sediment inorganic carbon deposition associated with calcifying organisms and dissolution, which may have a warming or cooling effect dependent on the formation processes of inorganic carbon in different coastal ecosystems, e.g. seagrasses and chenier plains, citing the relevant references.	1. 54-57
3. The third argument, that stocks are underreported because they do not include dead wood I think was also a bit weak. I never saw any citations or summary statistics indicating how many studies lacked dead wood, or how many included it. I know that there is guidance for accounting for dead wood in the IPCC wetland supplement guidance. I can also think of a few studies off the top of my head that do include it, although they tend to be tidal freshwater forested literature (Megonigal 1997; Krauss et al., 2018). Maybe there's a hole in the	The global syntheses to date do not account for dead biomass in ecosystem carbon stock, while our Supplementary Dataset covers individual studies of sediment OC stock. We combine the overall sediment OC stock (median value of individual sediment OC stock) with reported carbon stock of dead biomass and other components of carbon stock, to estimate ecosystem carbon stock in mangroves. Only 8.6% of references in our dataset reported dead wood. We	1. 60-61, Supplementary Dataset on downed/dead wood

mangrove literature I'm not aware of that could be communicated better than answered by this point.	have included these references in our dataset on dead wood biomass carbon.	
Overall comments		
1. The authors claim that saltmarshes and mangroves have different OM to OC ratios. If they think this is the result of a biological mechanism, I would like this explored more in the discussion. Also, could be that this is an artefact of the data synthesis. This could be addressed by trying a mixed-effects model, the same model as is done here with a 'study level' random effect.	We have provided a possible mechanism explaining the difference in relationships between mangrove and saltmarsh OC and LOI. The LOI data can have laboratory specific biases since our data are a combination of synthesised data from multiple sources and our field data. The individual study (i.e. 'study level') can be a random factor in the model, similar to Holmquist (2018). More importantly, however, the ranges of LOI and OC for individual studies of the two ecosystem types are also different without overlapping (Supplementary Information Table 1). So the random effect of individual data is already considered in the regression equations. From this point of view, the inclusion of 'study level' as a separate random effect could not explain the specific biases from different laboratories and is not considered in our revision.	1. 188-198
2. Figures – Figure 1. The log transformation makes the 95% CI's look like their exactly on the median line. There is missing symbology in the equations. The images are unnecessary. The top figure has log-transformed axes and the bottom they are linear. This is misleading and implies that the two relationships have more similar fits than they actually do. I recommend putting them on the same comparable axes and adding the 0.58 gOC per gOM line in for comparison.	We have used the same axes for both regression relationships but not in one figure since the large amount of data points will obscure which ones come from mangroves or saltmarshes. The missing symbol is due to transformation of word to pdf document in the online submission system and we will avoid the problem in the revised submission. We have also added the line with the slope of 0.58 (OC/OM) for comparison.	Fig. 1
3. I suggest a more thoughtful use of color. The color schemes for figures 2 and 4 are the same, but the colors mean different things. I suggest changing the palates in one or both.	We have used another colour scheme for Fig. 2.	Fig. 2

4. The histogram in fig. 2 doesn't really communicate the fact that there are significant differences in carbon stocks between latitudinal clusters to me. The variance in carbon stocks at different latitudinal gradients seems interesting and maybe deserves more attention in the main text.	The histogram in fig.2 (fig.2a) just shows the distribution of sediment carbon stock at various latitudinal ranges. The difference is shown in fig.2b and the K-W chi-square value is added. We have made it clear that the significant difference in carbon stocks can be observed in fig.2b rather than fig.2.	l. 104
5. SI table – There is a lot of unpublished data listed, and I would like to see any personal communications or submitter contact information cited in SI.	We have indicated the unpublished data coming from Atwood et al. (2017) in the Supplement Dataset.	Supplementary Dataset on carbon references
6. I suggest a column indicating the precision (number of decimal points) appropriate for the lat-long's.	We have added two columns indicating the precision for the latitudes and longitudes.	Supplementary Dataset on sites, cores and depths
7. I suggest machine-readable and analysis-ready datasets for everything they've produced in house (Wilson et al 2017; Culina et al 2018). I also suggest adopting the hierarchical set of site, core, depth series table format recommended by the Coastal Carbon Research Coordination Network (https://serc.si.edu/coastalcarbon/database-structure#site-level).	We have prepared the datasets that are ready for analysis. The dataset on sediment OC stock has been supplemented with the hierarchical set of site, core and depth series, including study and core ID, core elevation, salinity code, vegetation code, inundation code, core length, minimum and maximum core depth, and relative sea level rise over the late Holocene zones. We have also provided other Supplementary datasets.	Supplementary Datasets
Specific comments		
Line 62: Is the site, the core, or the depth interval the fundamental unit of the analysis? I think depth interval is best.	We have provided depth intervals in the Supplementary Dataset on sediment OC.	Supplementary Dataset on sites, cores and depths
Line 71: I'd prefer a more precise or quantitative phrase than 'clear cut'.	We have replaced the phrase but not a quantitative one because the polynomial and linear relationships cannot be compared quantitatively.	l. 88
Line 73: Somewhere in the results section it would be good to say how many significance tests were run, how independent they were. Consider running a Bonferroni correction to p-values to account for multiple testing.	We have run the Bonferroni correction to P values for multiple tests here and hereafter.	l. 89-90, l. 128-130, l. 328-330

Line 76-78: The claim is that 0.58 gOC per gOM is adopted by most studies. I haven't seen any evidence for this in the paper. In the SI it seems like a small subset of 13 studies. Also when reporting the regression equations, the authors should add the standard error of the parameter estimates. I think something more rigorous is needed than dropping a parameter because it is not significantly different from one. I suggest comparing a linear and polynomial versions of the model fit separately using Akaike's Information Criterion. Were salt marsh and mangrove relationships fit separately, or were they fit using the same model and an ecosystem type fixed effect?	We have modified the sentence to avoid misunderstanding. Standard errors of the parameter estimates have been added when reporting the regression equations. The polynomial term is significantly different from one and the exponent is not dropped after further updating the dataset on mangrove OC and LOI. So the linear relationship cannot be used to approximate the polynomial relationship for mangroves. We have clarified that the relationships for saltmarshes and mangroves were fitted separately.	Supplementary Information, text and Table 3, l. 96-98, l. 64
Line 80-89: How does this compare with Rogers et al 2019, who found that carbon stocks were higher in the regions where there was more historical isostatic sea-level rise forming accommodation space	The latitudinal change in sediment OC stocks is not comparable with the data in Rogers et al. 2019, but we supplement the variation of sediment OC stocks with relative sea level rise over the late Holocene zones, and found it consistent with the reference.	l. 111-116
Line 96: Would this still be significant after adjusting for multiple testing?	Yes. It is still significant after the Bonferroni correction.	l. 136
Line 102: I think this sentence should be in the abstract. Was it the correction of past studies using the 0.58 gOC per gOM or the switch from reporting means to medians which caused the 24% drop in total estimated stocks?	The information is shown in the abstract. The drop in the total estimated OC stock is mainly due to the switch from reporting means to medians, which is statistically more desirable to avoid the undue effect of extreme values. Other factors also result in the difference, including the different conversion factors used, the number of studies (147 studies in the previous synthesis and 235 studies in ours).	l. 16, l. 204-213
Materials and methods – I could use more details regarding the scenarios depicted in Fig. 4. I did not see them referenced in the main text at all and when I finished reading the paper I did	We have described in more detail about the different scenarios in Fig.4.	l. 338-341

not have a good idea as to where they came from and how they fit into the research questions.		
Reviewer #3		
Overall comments		
1. My primary criticism is that it was unclear whether the goal of this manuscript was to critique and suggest improvements for factors to convert loss-on-ignition (LOI) to organic carbon (OC) or to provide an updated global mangrove C stock estimate. In its current form, the manuscript attempts to do both, but the combination of these separate goals created many weaknesses and elements of confusion. As one example, it was confusing whether the updated global C stock estimates were a result of the different conversion factors, or the new data collected, or other reasons. I recommend splitting the manuscript into two.	We have reiterated in the ms that it is necessary to improve factors to convert OC from LOI due to (1) the wide use of conversion factors (e.g. 0.58) in individual case studies to estimate OC from LOI, and (2) the previous global estimates on sediment OC stock are based on conversion factors estimated from samples from single mangrove sites or non-mangrove habitats. This will partly, if not entirely, bias the estimate on global sediment OC stocks in mangroves. Thus, it is necessary to obtain a conversion factor based on global data before the global estimate of sediment OC stock may be improved. The difference between our estimated global C stock and previous estimates is attributed to (1) the corrected conversion factors; (2) new data collected; and (3) a full inventory of carbon pools, as explained in response to Reviewer #2's specific comment above. The messages are strongly interrelated and we therefore prefer to keep both objectives in one ms.	1. 37-44, 1. 204-213
2. I suggest adding: a clear and comprehensive review of conversion factors that have been developed in the literature. A similar review of how these conversion factors have been applied. A table summarizing these uses would be very helpful. This could be restricted to global C stock estimates, but could also be highly valuable if it covers C stock estimates at other scales. The use of conversion factors is far broader in scope than just for global C stock estimates.	We have added a table listing published conversion factors for coastal wetlands in the Supplement Information. Please also refer to our response to this point in the reply to Reviewer #2' comment above.	Supplementary Information Table 3, 1. 180-201

3. For global mangrove C carbon stock estimates, I suggest adding: A clear and comprehensive review of previous global mangrove C stock estimates and how and why they differ from the present study. A table would be useful.	We have added a table listing previous global mangrove C stock estimates, and compare them with our results.	Table 1
Additional comments		
The manuscript focuses on the role of coastal wetlands in mitigating climate change, but then criticizes previous studies for not including inorganic C in C stock estimates. Inorganic C is of course a component of the total C pool of a wetland, but is far less dynamic than organic C and probably plays little role in the ecosystem service of greenhouse gas mitigation. For this reason, it seems best to separate these pools when doing C stock estimates as a means to understand greenhouse gas mitigation.	We have highlighted the different nature of sediment OC and IC. Our result found that sediment IC is 14.6% of sediment OC stock, and is an important component of ecosystem carbon stock in mangroves. It may still have warming or cooling effect depending on the formation process (refer to our response to Reviewer #2's comment on argument 2). With other changes such as warming and ocean acidification affecting this component, we feel it necessary to include IC in our analysis.	l. 54-60, l. 151-153
Also related to greenhouse gas mitigation is the issue of human alterations. There is a substantial difference in terms of understanding climate forcing in quantifying C stocks and quantify C stock changes resulting from human activities. This separation is generally not made clear in the manuscript.	We have discussed the impact of climate on mangrove C stocks. Refer to our response to Reviewer #1's specific comment 9.	l. 231-237
The carbon stock estimate was only for mangroves, but the conversion factors were for both mangroves and marshes, which was confusing and not made clear in the paper.	We have further clarified this in the revised ms.	l. 64-65
There are quite a few comments such as the one on line 35 "but has been ignored in current estimates" without references. See lines 39-40 and 76 as other examples, but there are others as well. This would be remedied by have a more comprehensive analysis of conversion factors in the literature. The manuscript is written as if the 1.724 LOI/OC ratio is essentially the only one that has been used in the literature, but this is incorrect – there are many examples of other ratios that	This comment has been addressed along with this Reviewer's overall comment 2 and Reviewer #2's comment on argument 1. The different LOI/OC or OC/LOI ratios are compared with our study in a table, as stated above.	Supplementary Information Table 3

have been used. In fact, in a recent important publication this isn't the ratio that is suggested for use in mangroves (see Table 3.7 in Fourqurean et al. - 2014 - Coastal Blue Carbon Methods for Assessing Carbon). The authors may also consider if there is pertinent information in the recently published book A Blue Carbon Primer: The State of Coastal Wetland Carbon Science, Practice and Policy.		
The authors state on line 77 that the polynomial relationship was not statistically significant, but do not explain/defend why it was then used.	This point has been explained in response to Reviewer#1's specific comment 6.	l. 96-98
The manuscript is written as if the new data that were collected were global in nature, but only in the methods is it clarified that all of these data come from China (I believe). There is nothing wrong with adding data from China to this global database, but it should be explicit throughout the manuscript. It should also be discussed about whether these data fill a data gap—was China and/or these wetland types a data gap? Is this new database in any way weighted with the large number of samples from China?	We have further clarified this point in the ms to avoid confusion. We collected the new data from China as a supplement to the dataset to examine the relationship between LOI and OC, and sediment IC stock. Our field data from China (Hong Kong) covers both low and high mangrove sediments from scrub and forest (tall) mangroves, respectively. This has practical implications for the interpretation of the polynomial relationship found between mangrove OC and LOI. New sediment OC data are not only from China but also other regions of the world.	l. 282-284, l. 192-196
The use of the term “correct” should be replaced with something like “improve”. As scientists, we will never have the “correct” answer for a statistical population.	We have replaced ‘correct’ with ‘improve’ where suitable.	Title, l. 65, l. 165, 318-319, 322

Note: the line numbers listed in the above table are numbers under the ‘No Markup’ mode.

REVIEWERS' COMMENTS:

Reviewer #1 (Remarks to the Author):

The authors have diligently attended to most of the reviewer suggestions and queries.

The new table that compares global estimates should also include the source of area estimate of mangroves used to scale up the estimate. Using Giri et al. 2011 vs Hamilton and Casey, or other estimates is an important component of these calculations.

Using the phrase "warming or cooling" effect with respect to inorganic carbon is very unclear. The meaning should be made clearer.

Reviewer #2 (Remarks to the Author):

They authors addressed my concerns. I especially appreciate the effort to curate the supplemental data release for reuse. I scanned through them and didn't see anything that popped out.

A couple nitpicks though in the comments response that I think need a second look before it's ready for publication.

Comments:

Authors say, only 8.6% of the references reported dead wood. I think that should go into the text. Not essential, just a suggestion.

Authors say on line 96: that the term is still significant after the Bonferroni connection.

First, they say only three tests, but I see many more p-values listed.

The new significance threshold they use is 0.017.

The term is reported as significant at the 0.05 level.

What does this notation mean for discussing p-values? ' $p < .05$ '

Taking a second look at these figures I have two suggestions.

Fig 3: Consider putting units in legend

Typo in Fig. 4 Above ground biomass

Capitalize Belowground biomass to be consistent with other legend items.

Reviewer #1 (Remarks to the Author):

- 1. The authors have diligently attended to most of the reviewer suggestions and queries.**

Response: We thank the reviewer for acknowledging our effort to revise the ms.

- 2. The new table that compares global estimates should also include the source of area estimate of mangroves used to scale up the estimate. Using Giri et al. 2011 vs Hamilton and Casey, or other estimates is an important component of these calculations.**

Response: the new table estimates unit area carbon stocks (Mg ha^{-1}). Total carbon stock (Pg) has been estimated in the main text as unit area carbon stocks multiplied by mangrove area, which has included Giri et al. 2011 and other global estimates on mangrove area (1.174-5, 1. 218).

- 3. Using the phrase "warming or cooling" effect with respect to inorganic carbon is very unclear. The meaning should be made clearer.**

Answer: the warming or cooling effect describes the dissolution or release, respectively, of inorganic carbon during the transformation of different species of inorganic carbon.

Reviewer #2 (Remarks to the Author):

- 1. They authors addressed my concerns. I especially appreciate the effort to curate the supplemental data release for reuse. I scanned through them and didn't see anything that popped out.**

Response: We thank the reviewer for acknowledging our effort to provide the supplementary datasets.

A couple nitpicks though in the comments response that I think need a second look before it's ready for publication.

Comments:

- 1. Authors say, only 8.6% of the references reported dead wood. I think that should go into the text. Not essential, just a suggestion.**

Response: We have clarified this in the text by adding 17% of the references reporting dead biomass (including dead wood) (l. 84-5).

- 2. Authors say on line 96: that the term is still significant after the Bonferroni connection. First, they say only three tests, but I see many more p-values listed. The new significance threshold they use is 0.017. The term is reported as significant at the 0.05 level. What does this notation mean for discussing p-values? 'p<<.05'**

Response: The p values present in the paragraph are estimated for different relationships, i.e. mangrove and saltmarsh data, respectively. For mangrove data, three independent tests were run and the significance threshold is thus set as 0.017. Now the p values are all < 0.001 . Even much lower thresholds were set, the tests are still all significant. For the notation 'p<<.05', p is around 4.5×10^{-20} . We have replaced it with p<<0.001 (l. 98).

3. Taking a second look at these figures I have two suggestions. Fig 3: Consider putting units in legend. Typo in Fig. 4 Above ground biomass. Capitalize Belowground biomass to be consistent with other legend items.

Response: Units have been added to the legend of Fig. 3. Typos have been corrected for Fig. 4.

Note: the line numbers presented in our responses are the numbers in a clean version of the ms without tracked changes.